# Characterisation and Mechanical Modelling of Polyacrylonitrile-Based Nanocomposite Membranes Reinforced with Silica Nanoparticles

**DOI:** 10.3390/nano12213721

**Published:** 2022-10-23

**Authors:** Seren Acarer, İnci Pir, Mertol Tüfekci, Tuğba Erkoç, Vehbi Öztekin, Can Dikicioğlu, Güler Türkoğlu Demirkol, Sevgi Güneş Durak, Mehmet Şükrü Özçoban, Tuba Yelda Temelli Çoban, Selva Çavuş, Neşe Tüfekci

**Affiliations:** 1Department of Environmental Engineering, Faculty of Engineering, Istanbul University-Cerrahpasa, İstanbul 34320, Turkey; 2Faculty of Mechanical Engineering, Istanbul Technical University, İstanbul 34437, Turkey; 3South Kensington Campus, Department of Mechanical Engineering, Imperial College London, Exhibition Road, London SW7 2AZ, UK; 4Department of Chemical Engineering, Faculty of Engineering, Istanbul University-Cerrahpaşa, İstanbul 34320, Turkey; 5Department of Environmental Engineering, Faculty of Engineering-Architecture, Nevsehir Haci Bektas Veli University, Nevsehir 50300, Turkey; 6Faculty of Civil Engineering, Yıldız Technical University, İstanbul 34220, Turkey

**Keywords:** nanocomposite membrane, polyacrylonitrile, fumed silica, membrane characterisation, mechanical modelling

## Abstract

In this study, neat polyacrylonitrile (PAN) and fumed silica (FS)-doped PAN membranes (0.1, 0.5 and 1 wt% doped PAN/FS) are prepared using the phase inversion method and are characterised extensively. According to the Fourier Transform Infrared (FTIR) spectroscopy analysis, the addition of FS to the neat PAN membrane and the added amount changed the stresses in the membrane structure. The Scanning Electron Microscope (SEM) results show that the addition of FS increased the porosity of the membrane. The water content of all fabricated membranes varied between 50% and 88.8%, their porosity ranged between 62.1% and 90%, and the average pore size ranged between 20.1 and 21.8 nm. While the neat PAN membrane’s pure water flux is 299.8 L/m^2^ h, it increased by 26% with the addition of 0.5 wt% FS. Furthermore, thermal gravimetric analysis (TGA) and differential thermal analysis (DTA) techniques are used to investigate the membranes’ thermal properties. Finally, the mechanical characterisation of manufactured membranes is performed experimentally with tensile testing under dry and wet conditions. To be able to provide further explanation to the explored mechanics of the membranes, numerical methods, namely the finite element method and Mori–Tanaka mean-field homogenisation are performed. The mechanical characterisation results show that FS reinforcement increases the membrane rigidity and wet membranes exhibit more compliant behaviour compared to dry membranes.

## 1. Introduction

Membrane processes used for the supply of safe drinking water are advanced treatment technologies that overcome the shortcomings of traditional water treatment methods due to their advantages such as high separation performance, small footprint, no by-product, low cost and ease of use [1,2].

Polymeric membranes are more commonly preferred in membrane processes due to their low cost, ease of manufacture, more minor space requirements for installation, and controllable pore size [3,4]. Polyacrylonitrile (PAN) is widely used in the preparation of microfiltration and ultrafiltration membranes due to its commercial availability, low cost, thermal stability and resistance to various chemicals such as sodium hydroxide, sodium hypochlorite and chlorine [5,6,7,8]. Mixed Matrix Membranes (MMMs) are a new class of membranes that combine the superior properties of both polymeric and inorganic membranes [9,10]. By adding inorganic nanomaterials to the polymeric membrane matrix, problems such as flux reduction and fouling encountered in conventional membranes are overcome, as well as simultaneous improvement in membrane properties and performance [11,12,13,14,15].

In recent years, it is reported that the mechanical strength, flux performance, hydrophilicity, and resistance to fouling of the membranes increase with the incorporation of carbon-based nanomaterials [16,17,18] and nanometal oxides [8,19,20] into polymeric membranes. Many researchers think that the properties and performance of the membranes are improved by incorporating nano SiO_2_ into membranes made of different polymers such as polysulfone [21], polyethersulfone [22], cellulose acetate [20], polyvinylidene fluoride [23,24], and polyacrylonitrile [24]. Mavukkandy et al. reported that the water permeability of 3 wt% FS-doped PVDF membrane prepared by the phase inversion method increased by 140% compared to the pure PVDF membrane [23]. Similarly, Polisetti and Ray’s study reported that the pure water flux increased from 216 L/m^2^h to 384 L/m^2^h with the addition of 1% nano SiO_2_ to the neat PAN membrane [24]. Jin et al. showed that the tensile strength and surface hydrophilicity of the membrane increased as the amount of nano SiO_2_ in the cellulose acetate membrane rose from 0 to 6 wt%. [2].

Membranes are exposed to various loads during their service life. Therefore, it is necessary to study the mechanical properties of the membranes and obtain more rigid membrane properties to increase further the service life of the membranes [25]. For this reason, there are studies in which various particle reinforcements are applied to the membranes. Fumed silica (FS) can also be given as an example of these reinforcement materials. In addition, the membranes are mechanically modelled in order to predict and compare the service life of the membranes. Both experimental and numerical studies can be considered in the mechanical modelling of membranes [26].

There are various experimental methods in the literature for modelling membranes. The tensile test, dynamic mechanic analysis (DMA), and nano-indentation are examples of these methods [10,25,27,28,29,30,31]. In light of these experiments, the elasticity modulus values of the membranes can be obtained, the rigidity and service life of the manufactured membranes can be predicted, and the effects of the membrane reinforcement material on the mechanical properties can be examined. Hygrothermal properties of composites and membranes also affect membrane mechanical properties [32]. In the literature, experiments during different hygrothermal conditions are performed by various researchers. Lu et al. conducted tensile tests on dry and wet Polycaprolactone (PCL)/chitosan/polypyrrole (PPy) composite membranes. The results showed that dry membranes exhibit more rigid behaviour, and higher tensile strength values are recorded with dry membranes [33].

In addition, composite membranes can be modelled using numerical methods [34]. There are several methods and studies in the literature for investigating composite structures’ mechanical properties [35,36]. In the Mori–Tanaka homogenisation method, the mechanical properties of the materials that make the composite structure give the mechanical properties of the composite material with closed and analytical form equations [37]. Since the Mori–Tanaka homogenisation method is one of the most common and convenient methods in multi-scale modelling, it can be used as an example in composite materials [38,39]. The finite element (FE) method is another example of a multi-scale modelling approach to composite materials. In the FE method, a representative volume element (RVE) is generated and stress analysis is carried out on this RVE by applying the periodic boundary conditions (PBC) [36,39,40,41]. As a result, micromechanical behaviour and mechanical properties at the macro-level of composite materials are obtained.

This study explores the effect of FS reinforcement on the characteristics and mechanical properties of polyacrylonitrile-based membranes. To achieve this aim, the phase inversion method is used to manufacture PAN membranes with and without FS reinforcement. The prepared membranes are characterised using Fourier Transform Infrared (FTIR) Spectroscopy, Scanning Electron Microscopy (SEM) contact angle, water content, porosity, mean pore size, and pure water flow tests. Apart from these, the techniques of differential thermal analysis (DTA) and thermal gravimetric analysis (TGA) are used to characterise to identify the thermal properties of the membranes. Finally, tensile tests are conducted to characterise the mechanical behaviour of both reinforced and unreinforced membranes under wet and dry conditions. To understand more about the mechanics, two computational techniques, namely Mori–Tanaka mean-field homogenisation and finite element analyses, are also employed. Thus, a comprehensive characterisation/investigation is performed, allowing a more efficient and feasible design in the application of municipal and industrial wastewater treatment as well as drinking water treatment facilities. In other words, this study aims to serve the greater goal of finding the best membrane formation that is more permeable, more feasible, has greater mechanical strength, larger thermally operating spectrum, longer fouling time, and therefore longer operational life.

## 2. Materials and Methods

### 2.1. Materials

Polyacrylonitrile (PAN, M_w_ = 150,000 g/mol) and dimethyl sulfoxide (DMSO) are purchased from Sigma-Aldrich (St. Louis, MI, USA). Hydrophilic fumed silica (FS, HDK N20) in powder form is obtained from Wacker. The SiO_2_ content of FS is >99.8%. The diameters of the silica particles are known to be between 20 and 50 nm.

### 2.2. Preparation of PAN/FS Blend Membranes

PAN and PAN/FS blend flat sheet UF membranes are prepared by the phase inversion method [22,26,34]. Different amounts of FS (0.1–1 wt%) are added into DMSO and dispersed at 60 °C for 10 min. Then, 16 wt% PAN is added to the solution and mixed using a magnetic stirrer (WiseStir MSH-20A) for 24 h at 60 °C until a homogeneous mixture is formed. The weight ratios of the materials used in the preparation of the membranes are given in Table 1. Then, an ultrasonic bath (Weightlab Instruments, Istanbul, Turkey) is used for 30 min at room temperature to remove bubbles in the membrane solution. All membrane solutions are poured onto the glass plate using a 200 µm-thick casting blade (TQC Sheen, VF2170-261, Capelle aan den IJssel, Netherlands). The polymeric films on the glass plate are immersed in a coagulation bath containing non-solvent (distilled water). Membrane formation occurs due to the change between solvent–non-solvent, and the obtained membranes are thoroughly washed with distilled water. The membranes are kept in distilled water at 4 °C until their characterisation.

### 2.3. Characterisation of Membranes

#### 2.3.1. FTIR Analysis

Chemical groups on the surfaces of PAN and PAN/FS membranes are analysed using Fourier Transform Infrared (FTIR) Spectrometer (Perkin Elmer Spectrum 100, Waltham, MA, USA). The FTIR spectra of the membranes are recorded in the wavenumber range of 4000–650 cm^−1^.

#### 2.3.2. Thermal Characterisation

DTA-TGA apparatus (Shimadzu DTG-60) is used to determine the thermal character of FS and PAN/FS membranes at a heating rate of 10 °C min^−1^ between a temperature range of 25 °C and 700 °C under a nitrogen flow.

#### 2.3.3. SEM

The morphologies of the prepared membranes are investigated by Scanning Electron Microscope (SEM, FEI Quanta 250 FEG, Hillsboro, OR, USA) operating at 20 kV acceleration voltage. Membranes are coated with gold before SEM analysis to obtain surface and cross-sectional images of the membranes. SEM surface and cross-section views are examined at 10,000× and 1500× magnification, respectively.

#### 2.3.4. Contact Angle

The contact angle is measured using the contact angle goniometer (KSV CAM 101). The measurements are performed with distilled water at room temperature (25 °C) and repeated at least three times for each membrane to yield the average results and the standard deviations.

#### 2.3.5. Water Content

The water content of the membranes is determined by cutting three membrane samples of the same size from each membrane, and the water content results are averaged. First, dry weights of membrane samples, which are dried in an oven at 60 °C for 24 h, are determined with a precision balance to determine the water content of membranes. The membranes are then immersed in distilled water. The wet weights of the membranes are determined immediately after the excess water on the membranes is quickly removed with a drying paper. The water content of the membranes is calculated with the following equation:(1)$$\mathrm{Water}\;\mathrm{content}\;(\%)=\frac{W_{w}-W_{d}}{W_{w}}\times 100$$

$W_{w}$ and $W_{d}$ are membrane samples’ wet and dry weights (g), respectively.

#### 2.3.6. Porosity and Mean Pore Size

The porosity (*ε*) of the membranes is measured using the dry–wet weight method. The following equation is used to calculate the membrane porosity:(2)$$\varepsilon (\%)=\frac{W_{w}-W_{d}}{A\;l\;\rho \;}\times 100$$
where *W_w_* is the wet weight of the membrane (g); *W_d_* is the dry weight of the membrane (g); *A* is the membrane area (cm^2^); *l* is the membrane thickness (cm); and *ρ* is the density of the water (0.998 g/cm^3^).

The membrane’s mean pore size (r_m_) is calculated using the Guerout–Elford–Ferry equation below.
(3)$$r_{m}=\sqrt{\frac{\left(2.9-1.75\varepsilon \right)\times 8\eta lQ}{\varepsilon \times A\times \;\Delta P}}$$
where *ε* is membrane porosity; *η* is the viscosity of water (8.9 × 10^−4^ Pa.s); *l* is membrane thickness (m); *Q* is the volume of permeate water per unit time (m^3^/s); *A* is effective membrane area (m^2^); and Δ*P* operating pressure (0.3 MPa).

#### 2.3.7. Pure Water Flux Performance

Pure water flux tests of the membranes are carried out in a dead-end filtration cell (Tin Mühendislik) under 3 bar pressure using nitrogen gas. The active area of the membranes used in the filtration system is 19.62 cm^2^. The filtrate is collected in a beaker on a precision balance. The increase in the weight of the membrane upon soaking in water is plotted against time, from which the water flux into the membrane is calculated (L/m^2^ h). The pure water flux of the membranes is calculated with the following equation:(4)$$J=\frac{V}{A\times \Delta t}$$
where J is the membrane flux (L/m^2^ h); V is the permeate volume (L); A is the effective membrane area (m^2^); and t is the time (h).

### 2.4. Membrane Mechanics

Due to many parameters that affect the properties of polymer composites, it is essential to use simple but accurate material modelling techniques in polymer composite material modelling.

#### 2.4.1. Experimental Characterisation of Membrane Mechanics

The tensile test is one of the test methods for conducting mechanical properties of materials experimentally. With tensile tests, materials force-deformation data are obtained. With these data and stress and strain equations, the material’s stress–strain diagram can be drawn, and the material’s elasticity modulus, tensile modulus and elongation at break can be determined. These data are used to calculate the mechanical properties of the material which can be compared with other materials. In this study, the strain rate for quasi-static testing is determined to be less than 1% strain per minute. All kinds of membranes are tested both in wet and dry (24 h air drying at room temperature) conditions. The tensile test of the specimen can be seen in Figure 1. At the tips of the membrane specimens, aluminium plates are attached to avoid slipping at the clamps. Tensile tests of the membranes are conducted with Shimadzu AG-IS 50 kN universal test machine (Kyoto, Japan).

#### 2.4.2. Numerical Modelling of Membrane Mechanics

In order to investigate the mechanics of manufactured composite membranes, numerical modelling methods are used in this study (Python 3.10.6 and Abaqus 2020, Dassault Systems, Vélizy-Villacoublay, France). With numerical modelling methods, results and comparisons would be fast and effortless. Due to these conveniences, numerical modelling is one of the methods applied in the literature. In this study, the Mori–Tanaka Homogenisation method and the finite element (FE) method are used for the numerical modelling of composite membrane mechanics.

While conducting numerical analyses, the mechanical properties of the inclusions and neat membranes are determined, and then the mechanical properties of the composite membranes are calculated. The matrix material and inclusion material are assumed as linear isotropic elastic. The rigid bonding approach is used for the matrix–particle interfaces. Neat membranes are considered for the matrix phase and the influence of FS reinforcement is studied.

In the FE approach, an RVE model is constructed for analysis. The first step is to create a FE model of an RVE with randomly sized and positioned reinforcement particles. The nanoparticles are arranged in random positions and orientations as none of the particles intersects, or the RVE boundaries and RVE geometry is generated [42,43,44]. This RVE is subjected to the uniaxial tensile strain of 0.03 on one surface with PBC. Effective mechanical properties and stresses on a small scale can be characterised by uniaxial tension. Figure 2 shows a sample distribution of FS particles inside the RVE.

## 3. Results

### 3.1. FTIR

FTIR spectra of PAN and PAN/FS membranes are given in Figure 3a,b. The characteristic peak of the nitrile group (-C≡N stretching vibration) in the PAN structure is observed for all membranes at 2244, 2243, 2244, 2244 cm^−1^ [45,46,47,48]. The bands at 2937, 2939, 2932, 2933 cm^−1^ and 1453, 1454, 1454, 1454 cm^−1^ are assigned to the -CH_2_- vibration peak [46] and the bending of the C-H bond in CH_2_ [47,48], respectively. Figure 3b presents the FTIR spectra of samples in the range of 650–1200 cm^−1^, where spectral differences can be observed. The significant bands present at 791, 791, 793 cm^−1^ are assigned to the Si–O deformation [49], and the bands at around 1100 cm^−1^ can be due to the asymmetrical stretching vibration of Si-O-Si [49,50], confirming the presence of FS in the membrane. The band at 1074 cm^−1^ is attributed to S=O groups of DMSO solvent [51]. The peak intensity at 1100 cm^−1^ increases as the FS content of the membrane increases (Figure 3b). It is reported that the peak intensities of PAN/silica nanofibers are higher at 1100 cm^−1^ than the corresponding pure polymer [52].

The peak intensities of the bands at 2937, 2244 and 1454 cm^−1^ in the PAN membrane decrease with the addition of FS, which probably can be ascribed to the cyclisation and dehydrogenation [51]. A band at about 1245 cm^−1^ [47] seems to appear in all four spectra as a shoulder. Si-OH stretching (between 3100 and 3700 cm^−1^) broadens when FS contents in the PAN membrane increase. It is stated that interactions may be formed between the silanol groups in silica and the PAN’s nitrogen atom [52]. 

### 3.2. Thermal Characteristics

TGA analysis is conducted on the membranes to determine and compare their thermal properties. When PAN is heated, chemical reactions such as cyclisation, degradation, and crosslinking can occur. Both thermal analysis conditions (polymer mass, atmosphere and heating rate) and filling material properties significantly affect the PAN-based membranes’ thermal behaviour [53]. It is reported that the thermal degradation process initiates a temperature lower than 300 °C in airflow while it starts above 300 °C in a nitrogen environment. Most nitriles are involved in cyclisation and trimerisation reactions at temperatures below 400 °C [54]. Since nitrile groups have a high dipole moment, internal chains are powerful, and interactions occur via secondary bonds in PAN [53].

TGA and DTA analysis results of PAN, PAN-FS0.1, PAN-FS0.5, PAN-FS1, and FS are shown in Figure 4. In the PAN membrane, the initiation temperature of the DTA exothermic peak is around 240 °C and the maximum peak is observed at 317 °C, which can be attributed to the cyclisation of nitrile (Figure 4a). The weight losses of PAN are 15.8, 26.4, 65.6 and 92.9% at 400 °C, 500 °C, 600 °C and 700 °C, respectively, as shown in Table 2.

When the TGA curve of FS is evaluated (Figure 4b), it is observed that there is no weight loss during the heating temperature between 25 °C and 700 °C under a nitrogen atmosphere and FS shows thermal stability at even high temperatures. It is previously reported that FS exhibits a superior thermal resistance [55].

The weight loss (%) of membranes at different temperatures is presented in Table 2. When the TGA curves of all polymers are analysed, no significant weight loss is observed up to 300 °C. Faraji et al. investigated the thermal behaviour of PAN nanofibers and stated that weight loss at up to 300 °C is very low due to the samples’ humidity [56]. The PAN-FS0.1 membrane exhibits better thermal resistance than the other counterparts (Figure 4c). This behaviour is evident at 600 °C. It is stated that using a suitable FS content can improve the flame retardant feature [57]. It is reported that the thermal stability of poly(lactic acid) FS nanocomposites is better than its corresponding pristine polymer [58].

The TGA curve of the PAN-FS0.1 membrane gives about 11.3% residual after reaching 700 °C but the residual is 3.2% for the PAN-FS1 membrane. It is seen that as the FS content of the membrane increases, the thermal stability decreases. In the study by Ye et al., the synergistic effect of FS is investigated for the polypropylene/NP phosphorus–nitrogen compound blend. It is concluded that an appropriate amount of FS is required to determine its synergist effect since excessive contents of FS can reduce the thermal stability of blends [57]. In our study, this synergistic effect can be attributed to the PAN-FS0.1 membrane with a suitable FS content.

In the current study, while the initiation temperature of the exothermic peak is around 250 °C and the maximum peak is observed at 328 °C for PAN-FS0.1, the initiation temperature is 215 °C and the maximum peak is determined at 323 °C for PAN-FS0.5 owing to the cyclisation reactions, as shown in DTA. When the FS amount increases, the initiation temperature of the exothermic peak in the PAN-FS1 membrane increases to 262 °C. In this case, when the reaction rate and the heat released are high, the polymer chains are broken down, producing volatile particles that cause weight loss [59]. Owing to the radical nature of nitrile cyclisation in the PAN-FS1 membrane, it initiates at a higher temperature and possesses a high reaction rate compared to other membranes, resulting in higher weight loss for PAN-FS1.

### 3.3. Membrane Morphology

Membranes are examined by SEM to determine the effect of FS on PES membrane morphology. Figure 5 shows SEM surface and cross-sectional views of PAN membranes with different FS content. The incorporation of FS into the neat PAN membrane increases the membrane surface porosity. The increase in porosity is due to the hydrophilic FS increasing the exchange rate between solvent and water during the phase inversion [21]. The incorporation of FS into the membrane also caused the internal structure to change. In comparison, the neat PAN membrane exhibits a porous structure extending from the surface to the bottom, the 0.1–1 wt% FS-doped PAN membranes show a dense skin layer on top, finger-like pores in the middle layer, and horizontal macro voids on the bottom layer. At low FS loadings to the membrane (0.1 wt%), a large number of thin finger-like spaces are formed in the middle of the membrane. When the FS content increases to 0.5 wt%, the size and number of pores formed on the surface, the size of finger-like pores in the middle layer, and the size of the macro voids in the bottom layer increase compared to the membrane containing 0.1 wt% FS. However, the greater amounts of FS addition (1 wt%) cause the size of the pores formed on the membrane surface and interior structure to decrease significantly and the membrane interior structure to become very dense. The highly dense structure of the 1% FS-doped membrane compared to other membranes can be attributed to the high FS content, significantly increasing the viscosity of the membrane casting solution [21,60]. In other words, the exchange rate between solvent and non-solvent during phase inversion decreases due to high viscosity and a denser membrane structure is formed.

### 3.4. Membrane Hydrophilicity

Using membranes with high surface hydrophilicity in water treatment is ideal for increasing flux performance and anti-fouling properties. The high surface hydrophilicity of the membranes is expressed by the low contact angle. Another parameter indicative of membrane hydrophilicity is the water content of the membranes. Figure 6 shows the contact angle and water content results of membranes. It is determined as the neat PAN membrane with the most hydrophobic surface with the highest contact angle (67.5°). As the FS addition increases, the surface hydrophilicity of the membranes increases, and the contact angle decreases to 36°. The hydrophilicity of nanocomposite PAN/FS membranes increases significantly due to the hydrophilic hydroxyl (-OH) groups in the structure of FS.

The water content (50%) of the pure PAN membrane increases until the addition of 0.5 wt% FS (88.8%) and then decreases. The porosity and pore size of the membrane surface and interior structure increases until the addition of 0.5 wt% FS enabled more water to be retained in the membrane structure. However, high FS loadings reduce the pore size and porosity in the surface and inner structure of the nanocomposite membrane, making it denser and resulting in less water absorption.

### 3.5. Porosity and Mean Pore Size

Figure 7 shows the porosity and average pore size results of the PAN/FS membranes. The mean pore sizes of the PAN, PAN-FS0.1, PAN-FS0.5 and PAN-FS1 membranes are determined as 21.3 ± 0.9 nm, 21.7 ± 0.6 nm, 21.8 ± 0.5 nm, and 20.1 ± 0.4 nm, respectively. The mean pore sizes of the prepared membranes vary in ca. 20–22 nm and these pore sizes correspond to the pore size range of ultrafiltration (UF) membranes [61]. Although the mean pore size of the membranes increases to 0.5% FS additive, the pore size decreases under excessive FS loading. The pore size decreases due to pore blocking as well as the higher FS loading increases the viscosity.

The porosity of the membranes varies between 62.1% and 90%. The acceleration of liquid–liquid demixing during phase inversion of hydrophilic FS causes an increase in membrane porosity. However, the porosity increases up to the addition of 0.5 wt% of FS and decreases at the addition of 1% by weight of FS. The high amount of FS in the membrane casting solution increases the viscosity of the solution and decelerates the liquid–liquid demixing during phase inversion, resulting in a decrease in porosity.

### 3.6. Pure Water Flux Performance

The pure water flux performance of the membranes under 3 bar pressure is given in Figure 8. The neat PAN membrane obtains the lowest pure water flux performance with a value of 299.8 L/m^2^ h. The pure water flux of the PAN membrane increases by 26% with 0.5% FS contribution and increases to 378.4 L/m^2^ h. The fact that hydrophilic FS increases the membrane surface hydrophilicity and porosity improves the pure water flux performance of nanocomposite membranes.

The pure water flux performance of 1% FS-doped nanocomposite membrane is lower than 0.5% FS-doped nanocomposite membrane. Due to the high FS loading making the membrane’s inner structure denser and reducing the mean pore size of the membrane, the passage of pure water through the membrane becomes difficult and the flux performance decreases. However, the pure water flux performance of the 1% FS-doped PAN membrane is 18.6% higher than the pure PAN membrane. According to this study, pure water flux test results show that the amount of FS in the membrane should be kept at 0.5% to obtain maximum water flux performance.

### 3.7. Membrane Mechanics

In this study, the mechanics of membranes are investigated using experimental and numerical methods. The tensile tests are performed at least three times for each parameter in order to ensure consistency. Accordingly, the results from the tensile test and the numerical methods are presented in this section. The tensile properties obtained from the experimental process are given in Table 3. When the results obtained from the tensile tests are investigated, it is seen that the FS addition improves the stiffness and strength properties of the membranes. While considering tensile test results for wet membranes, it is observed that the modulus of elasticity rises by 5.54%, 8.34% and 13.8% for 0.1, 0.5 and 1 wt% FS-doped membrane, respectively, compared to the neat membranes. For the dry membrane case, these increases in elasticity modulus are calculated as 4.19%, 5.61%, and 8.08%, respectively. Based on these results, the increase in elasticity modulus due to the FS additive is not linear and the growth depends on both the materials’ hygrothermal conditions and the weight percentage of the reinforcement material.

Additionally, when samples’ tensile properties are investigated, elongation at break increases with the increasing mass fraction of FS. It can be said that FS-reinforced membrane samples show more ductile behaviour in comparison to the neat PAN membrane as expected since FS is known to toughen the polymeric matrix materials [62]. When a membrane’s elongation at break value increments are evaluated, the 1 wt% FS-doped dry membrane has the highest increase with a 53.4% change compared to the neat membrane. The most ductile membrane is found as the 1 wt% FS-doped wet membrane with an elongation of 0.1309 at the break.

In Figure 9, the stress–strain diagrams of the manufactured membranes are given (wet/dry). From these diagrams, it can be claimed that wet/dry conditioned membranes possess different mechanical characteristics. Elongation at break values of wet membranes is almost ten times higher than dry membranes. Additionally, wet membranes’ tensile strength and elasticity modulus values are higher than dry membranes. When tensile strength values of FS-doped wet and dry membranes are compared, FS-doped dry membranes’ changes in tensile strength are greater than wet membranes. The effects of FS addition can be seen in dry membranes more significantly. It is found that the wet samples are more compliant but more ductile than the dry membranes. This is due to the change of the polymeric matrix materials under different hygrothermal conditions. These results are also compatible with the literature [33,63,64,65].

As another part of this study and these results, the comparison diagram of experimentally determined elastic moduli and numerically calculated elastic moduli of membranes are given in Table 4 and Figure 10.

Homogenised mechanical properties of composite membranes are calculated using the Mori–Tanaka homogenisation method and FE analysis. Predicted elasticity modulus values of each membrane are given in Table 4 and Figure 10. To ensure consistent and accurate results, the analyses are repeated five times for each material set.

From the numerical modelling results, it is possible to observe the effect of the FS additive on the pure membrane. In this case, it is seen that the FS-reinforced membranes show stiffer behaviour and the results agree with the tensile test results. In addition, when the wet–dry condition is compared, similar results are obtained from the experimental results. Comparing the results of the Mori–Tanaka homogenisation method to the FE results, it is seen that the stiffness values calculated with the FE are greater and the error rates are lower. This is an expected outcome since FE conducts a more detailed/less simplified stress analysis. Therefore, FE is more successful in terms of representing the effects of the reinforcing phases on the mechanical properties of the composites in general. The maximum error between the numerical and the experimental results is around 11% with the Mori–Tanaka homogenisation method.

A view of the FE stress analysis results of an RVE reinforced with FS is presented in Figure 11. Based on the FE analysis results, maximum stress is observed in the inclusions. It can be pointed out that the reinforcement material significantly affects the load-carrying capacity of the composite. Apart from that, around the FS particles, more significant stresses are visible, leading to local plastic deformations. These stress concentrations are also affected by the distance between inclusions. When the distance between inclusions decreases, the interaction between particulates increases. These interactions show the importance of the uniform distribution of reinforcement material in the matrix phase.

## 4. Conclusions

In this study, FS-doped PAN membranes are manufactured and characterised using several experimental procedures aiming to determine membranes’ thermal and mechanical properties to contribute to the membrane selection/tailoring process for the optimum membrane combination for using a water treatment system.

The average pore size results of the membranes revealed that all manufactured membranes are categorisable as UF membranes. The more FS is added to the PAN membrane, the more the membrane surface porosity rises. SEM cross-sectional images of the membranes show that the FS additive changed the internal structure of the PAN membrane and caused differences in the shape and number of finger-like voids. The results also suggest that even extremely low amounts of FS presence (0.1% by weight) could alter the membrane morphology.

With the addition of FS up to 0.5%, the water content of the PAN membrane increases and then declines. The parameters that are closely related to the water content are the porosity and contact angle. The highly dense (less porous) internal structure of the PAN-FS1 membrane causes less water content. Although the flux decreases with more FS present in the membrane, the pure water flux of 1% nanoparticle-doped PAN membrane is found to be higher than the neat PAN membrane.

With thermal gravimetric analysis, each membrane combination’s weight loss at several temperatures is evaluated and the results are listed in the relevant section. Thermal characterisation results show that the highest weight loss is observed for 1% FS-reinforced PAN membrane. Additionally, the 1% FS-doped PAN membrane has the highest reaction rate compared to other membrane samples.

In the mechanical characterisation part, experimental and numerical approaches are used and modulus of elasticity values are obtained for each membrane sample. In the experimental process, tensile tests are conducted; in the numerical analyses, the Mori–Tanaka homogenisation and FE analyses are performed. It is found that the membrane becomes stiffer with FS reinforcement and the 1% FS-reinforced PAN membrane is the most suitable membrane in terms of mechanical performance.

Among all the polymer-based flat sheet water treatment membranes produced, the 0.5 wt% FS-doped PAN membrane is found to be suitable for practical applications due to its highest flux performance and high hydrophilicity (surface wettability and water content). Although membranes become stiffer with FS reinforcement when the results are evaluated in terms of practical applications, the 0.5 wt% FS-doped membrane is more suitable for practical use.

## Figures and Tables

**Figure 1 nanomaterials-12-03721-f001:**
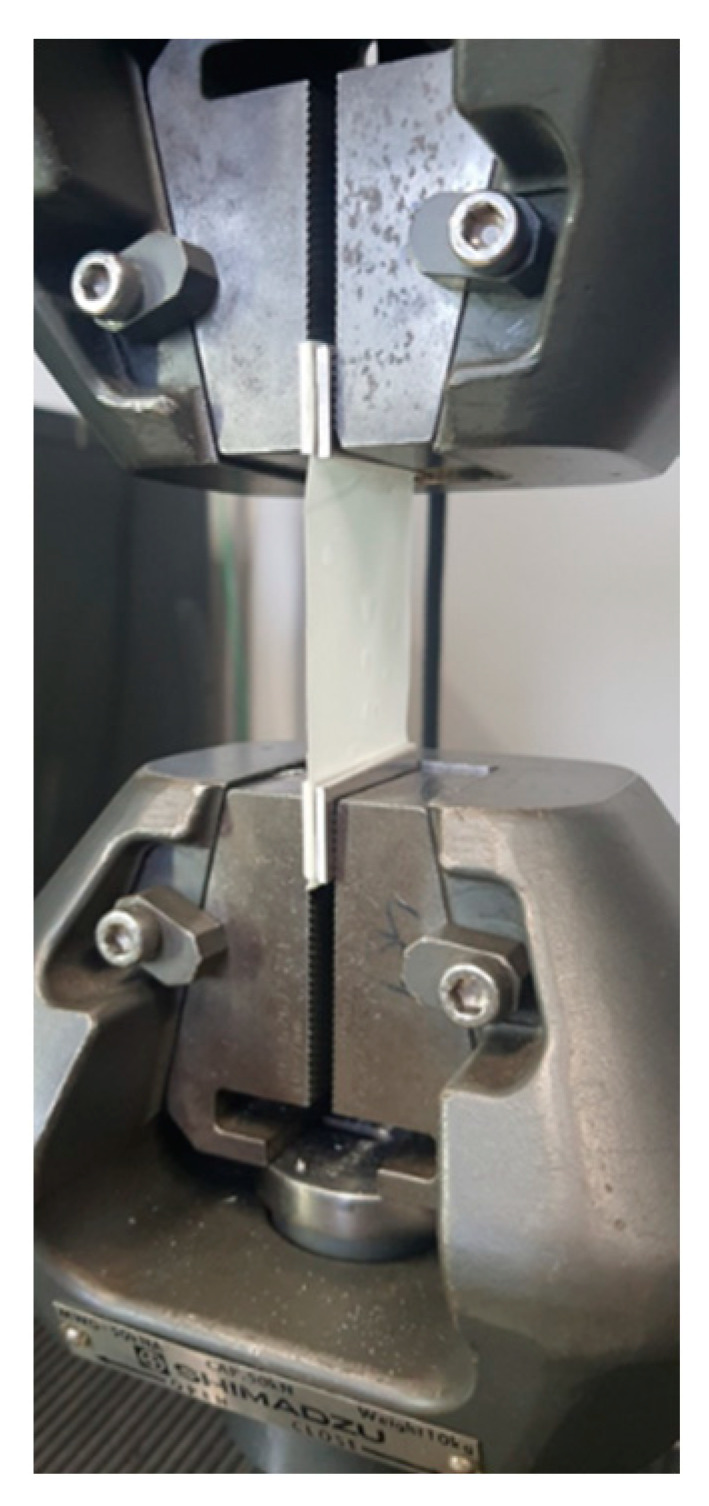
Tensile test specimen on the test machine.

**Figure 2 nanomaterials-12-03721-f002:**
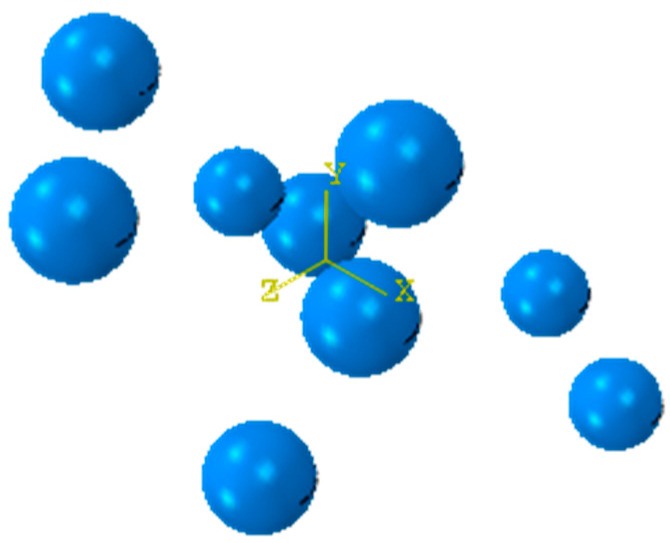
Distribution of reinforcing material in the RVE.

**Figure 3 nanomaterials-12-03721-f003:**
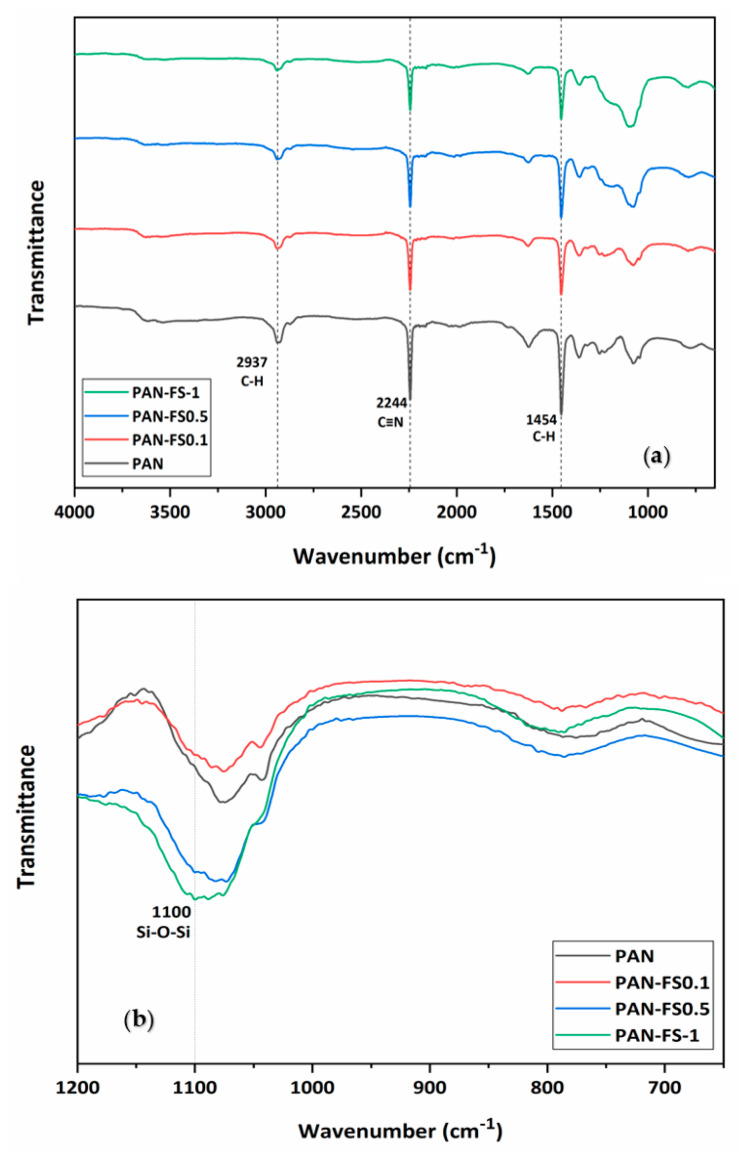
The FTIR spectra of the membranes are (**a**) 4000–650 cm^−1^ and (**b**) 1200–650 cm^−1^ wavenumber range.

**Figure 4 nanomaterials-12-03721-f004:**
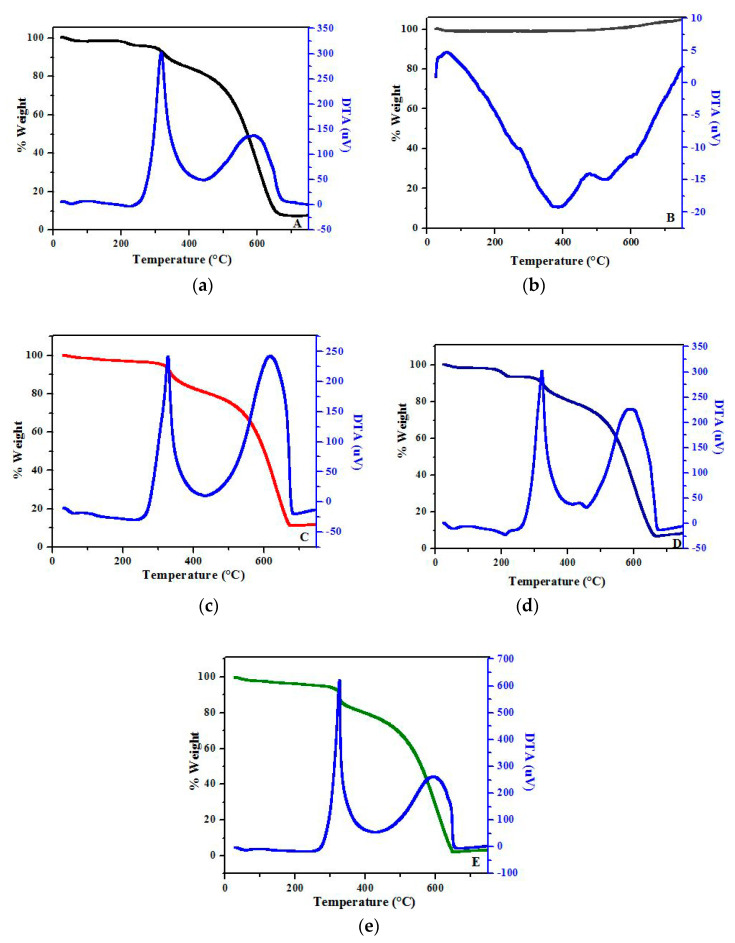
TGA and DTA curves of (**a**) PAN, (**b**) FS, (**c**) PAN-FS0.1, (**d**) PAN-FS0.5 and (**e**) PAN-FS1.

**Figure 5 nanomaterials-12-03721-f005:**
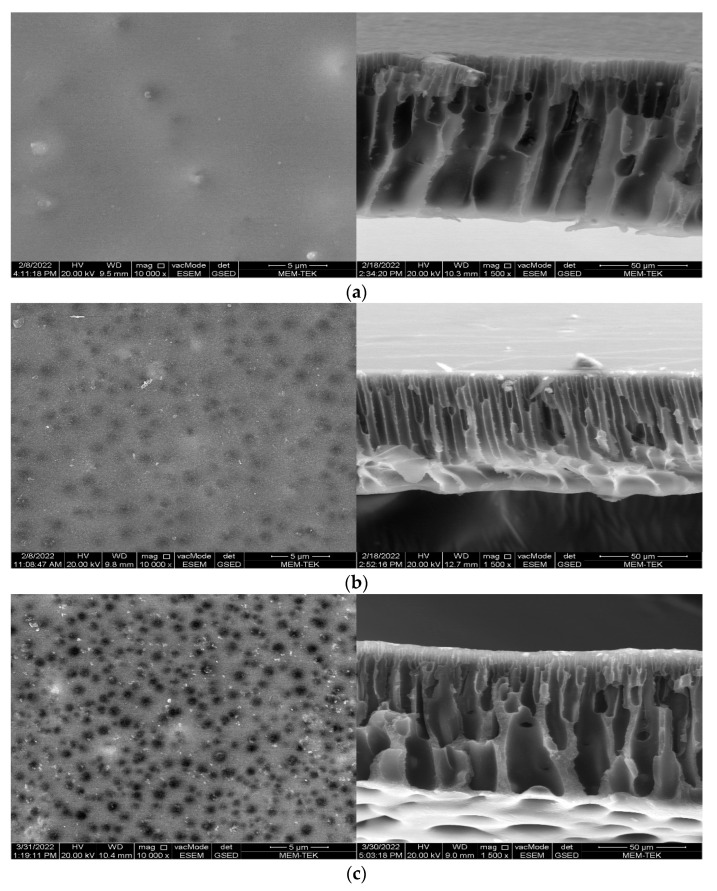
SEM surface and cross-section views (**a**) PAN, (**b**) 0.1% FS-doped PAN, (**c**) 0.5% FS-doped PAN and (**d**) 1% FS-doped PAN.

**Figure 6 nanomaterials-12-03721-f006:**
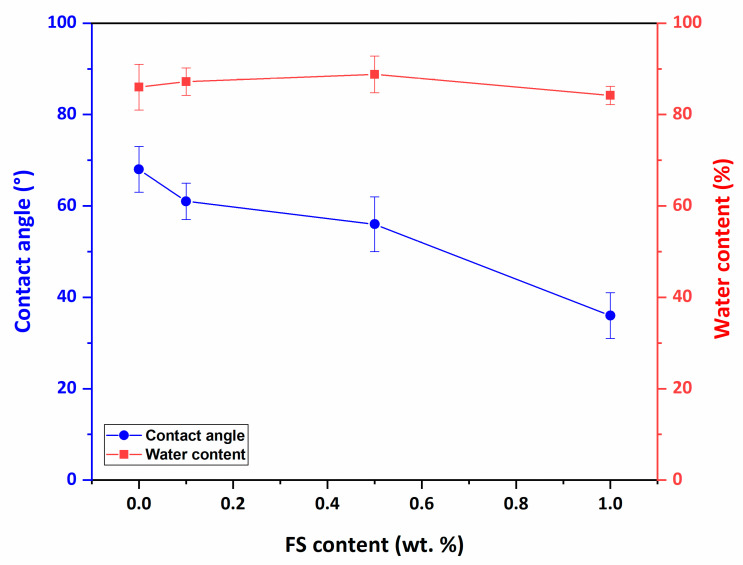
Contact angle and water content of PAN/FS membranes.

**Figure 7 nanomaterials-12-03721-f007:**
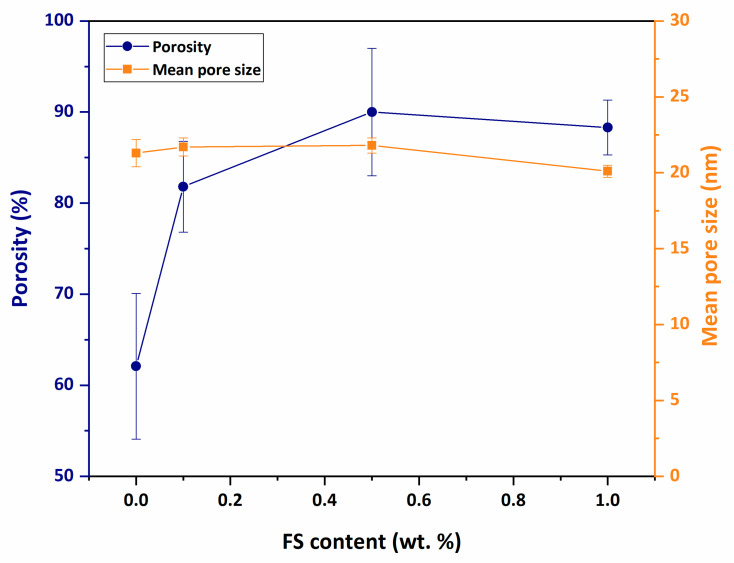
Porosity and mean pore size of PAN/FS membranes.

**Figure 8 nanomaterials-12-03721-f008:**
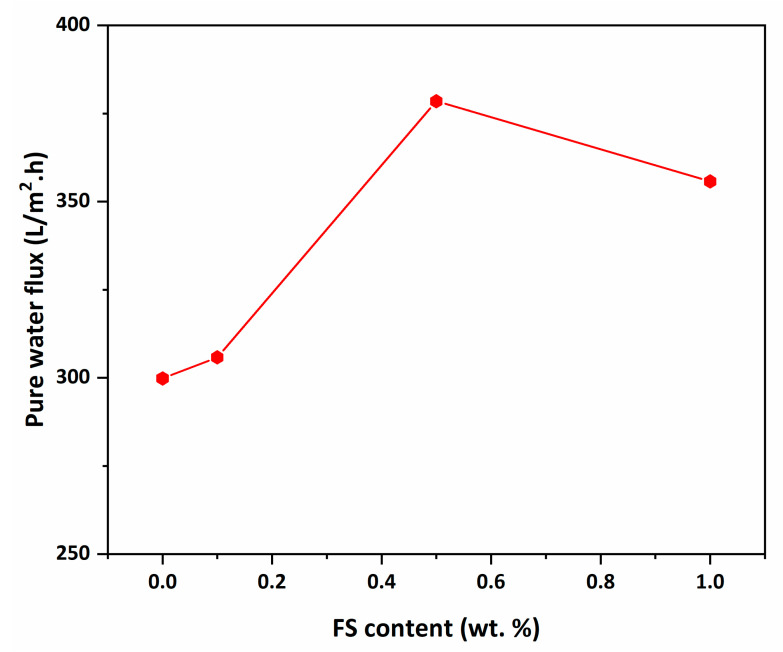
Pure water flux of PAN/FS membranes under 3 bar pressure.

**Figure 9 nanomaterials-12-03721-f009:**
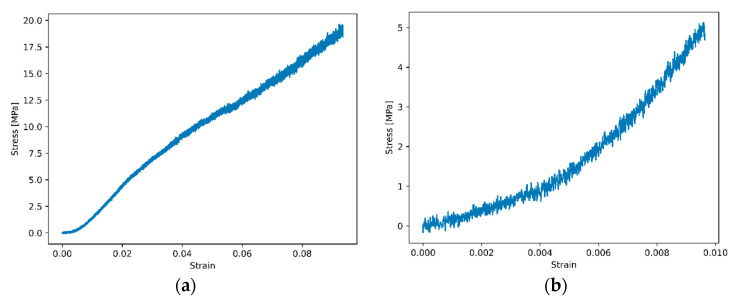
Stress–strain diagrams of (**a**) wet PAN-DMSO membrane and (**b**) dry PAN-DMSO membrane.

**Figure 10 nanomaterials-12-03721-f010:**
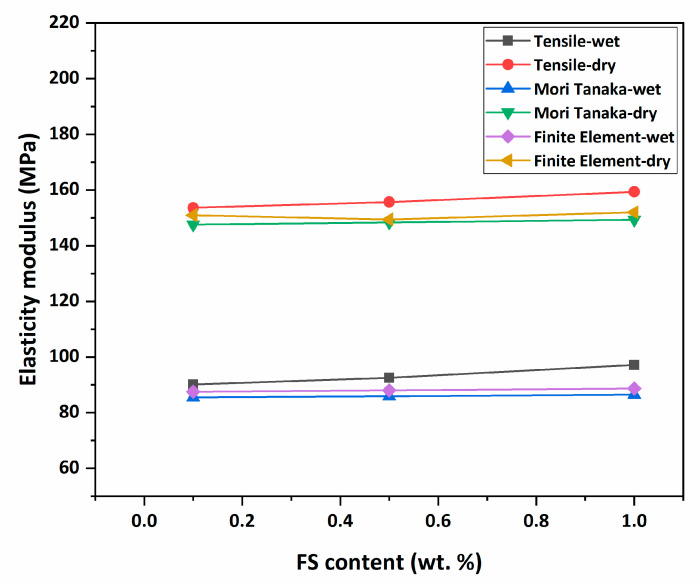
Elasticity modulus change of samples.

**Figure 11 nanomaterials-12-03721-f011:**
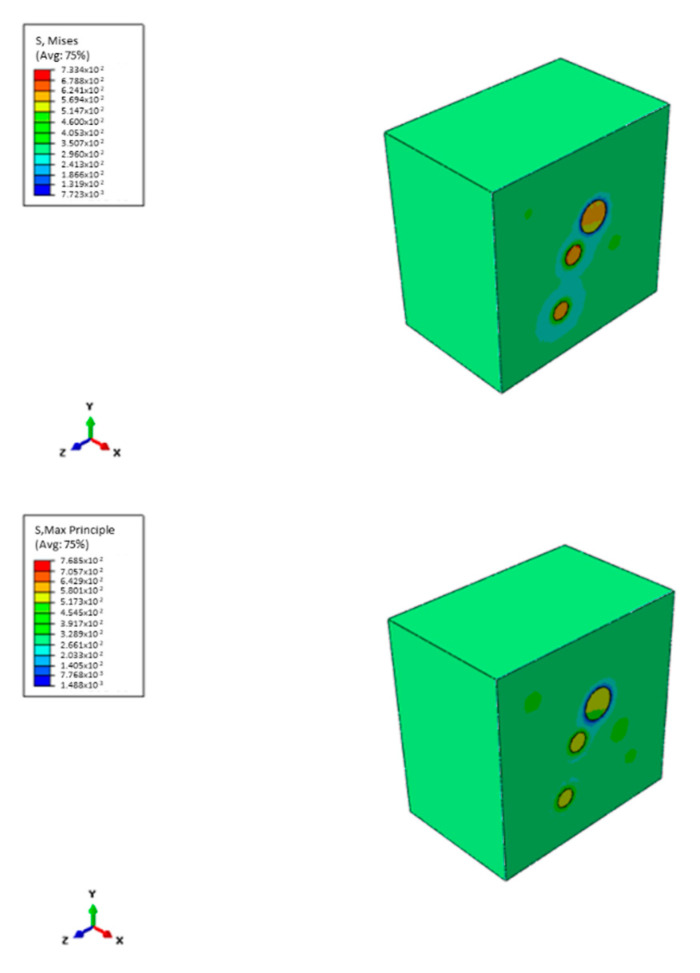
Finite element stress analysis of samples.

**Table 1 nanomaterials-12-03721-t001:** Composition of the prepared flat sheet membranes.

Membrane	PAN (wt%)	DMSO (wt%)	FS (wt%)
PAN	16	84	-
PAN-FS0.1	16	83.9	0.1
PAN-FS0.5	16	83.5	0.5
PAN-FS1	16	83	1

**Table 2 nanomaterials-12-03721-t002:** Weight loss percentages of prepared membranes for various temperatures.

Weight Loss Percentages							
Membrane Type	100 °C	200 °C	300 °C	400 °C	500 °C	600 °C	700 °C
PAN	2.1	2.2	5.4	15.8	26.4	65.6	92.9
PAN-FS0.1	1.4	2.8	4.2	17.1	24.2	49.2	88.7
PAN-FS0.5	1.9	4.2	7.6	19.6	28.6	65.0	92.9
PAN-FS1	2.0	3.3	5.3	19.7	31.1	70.8	96.8

**Table 3 nanomaterials-12-03721-t003:** Tensile properties of manufactured membranes.

		Elasticity Modulus (MPa)	Tensile Strength (MPa)	Elongation at Break
DMSO	Wet	85.393	19.626	0.0936
DMSO	Dry	147.43	5.1256	0.0096
0.1 g FS	Wet	90.13	20.1652	0.1192
0.1 g FS	Dry	153.6	6.1119	0.0101
0.5 g FS	Wet	92.52	22.0376	0.1257
0.5 g FS	Dry	155.7	7.2553	0.0124
1 g FS	Wet	97.18	23.54	0.1309
1 g FS	Dry	159.34	7.8926	0.0147

**Table 4 nanomaterials-12-03721-t004:** Elasticity modulus of each membrane combination.

		Experimental [MPa]	Mori–Tanaka [MPa]	Error (%)	FE [MPa]	Error (%)
DMSO	Wet	85.393				
DMSO	Dry	147.43				
0.1 g FS	Wet	90.13	85.501	5.140	87.563	2.852
0.1 g FS	Dry	153.6	147.62	3.893	150.951	1.725
0.5 g FS	Wet	92.52	85.938	7.114	88.004	4.881
0.5 g FS	Dry	155.7	148.37	4.708	149.446	4.017
1 g FS	Wet	97.18	86.488	11.002	88.713	8.713
1 g FS	Dry	159.34	149.32	6.288	151.976	4.622

## Data Availability

Not applicable.

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
