# Peer review of "Characterisation and Mechanical Modelling of Polyacrylonitrile-Based Nanocomposite Membranes Reinforced with Silica Nanoparticles"

_nanomaterials, 2022, doi:10.3390/nano12213721_

Round 1
Reviewer 1 Report
The manuscript deals to investigate the effect of silica nanoparticle addition on characteristics and mechanical strength of PAN membranes. The subject is interesting; it is related to the profile of the journal. The experimental design is appropriate, several up-to-date methods and equipment were used to prove the results. The manuscript is well written, clear, and understandable, figures are nice and supporting. The English of the manuscript seems to be good and understandable. (The referee is not a native English speaker).
Comments:
- The Title is informative.
- The Abstract reflects the approach of the study, nevertheless, the main results of the work also shoulb be presented here.
- The section Introduction presents the important points of the topic, contains references related to the work, and reveal the importance and novelty of the work.
- The section Experimental. The experimental design is appropriate; it is generally well described.
- In the section Results and discussion, authors describe the results shown in the corresponding figures, and tables. The figures are nice and informative.
- The Conclusions section well summarizes the result of the work, however, practical applicability or importance also should be presented.
- Conclusion and recommendation: This manuscript is recommended for publication after minor revision.
Reviewer 2 Report
See attached file

Reviewer 3 Report
The paper is moderately interesting it should be improved according to following lines:
The author should provide some quantitative information in the abstract section.
In section 2.2. Preparation of PAN/FS blend membranes, the author should provide reference.
The author should provide image quality in Fig. 3
FTIR spectra of PAN and PAN/FS membranes are given in Figures 3a and b. The characteristic peak of the nitrile group (-C≡N stretching vibration) in PAN structure is observed for all membranes at 2244/2243/2244/2244 cm-1 [46–49]. The bands at 2938/2939/2932/2933 cm-1 and 1453/1454/1454/1454 cm-1 are assigned to -CH2- vibration peak [47] and bending of C-H bond in CH2 [48,49], respectively. The author should change the text 2244, 2243, 2244, 2244 cm-1 instead of 2244/2243/2244/2244 cm-1 and also others.
XRD data should provide by the author for the confirmation of crystallinity nature.
Conclusion should be revised to show the outstanding point of this work.
Round 2
Reviewer 2 Report
Three more comments:
1. Line 42: SiO2 is a chemical formula, therefore should be removed fron the abbreviations list.
2. Line 322: a) Should be "were" instead of "is". b) PES should be added to the abbreviations list.
3. Line 376: The range of the pore-size can be written as ca. 20-22nm. The errors have already given before.
Author Response
We thank the referee for the guiding suggestions and corrections. The corrections were done and were listed below. All the relevant changes are highlighted in the manuscript.
1. Line 42: SiO2 is a chemical formula, therefore should be removed fron the abbreviations list.
Revised
2. Line 322: a) Should be "were" instead of "is". b) PES should be added to the abbreviations list.
Revised
3. Line 376: The range of the pore-size can be written as ca. 20-22nm. The errors have already given before.
Revised